# Dimensionality Reduction and Electrode Arrangement Optimization for an Electric Field Source Seeking Surgical Navigation Method

**DOI:** 10.3390/s25051378

**Published:** 2025-02-24

**Authors:** Yuxin Fang, Fan Yang, Wei He, Xing Li, Xinheng Li

**Affiliations:** School of Electrical Engineering, Chongqing University, Chongqing 400044, China; drfangyuxin@163.com (Y.F.); 20144294@cqu.edu.cn (F.Y.); cqulixing@cqu.edu.cn (X.L.); lixinheng@stu.cqu.edu.cn (X.L.)

**Keywords:** electrode arrangement, surgical navigation, electric field, biomedical sensors

## Abstract

This study proposes a Dimensionality Reduction Electric Field Source Seeking (EFSS) method for real-time, high-precision navigation in intracranial puncture surgeries. The method integrates internal localization electrodes and external potential measurement electrodes to minimize surgical trauma while ensuring the accurate localization and guidance of surgical instruments. To optimize the electrode arrangement, two evaluation metrics—Mean Response Coefficient (MRC) and MRC-mean—were introduced. The simulation results demonstrated the effectiveness of these metrics, with the optimal arrangement achieving an average localization error below 2 mm and a 56% reduction in error after optimization. Experimental validation was conducted using a brain model with conductivity properties similar to those of human tissue. Localization experiments confirmed the robustness and accuracy of the EFSS method, with all results showing consistent repeatability and monotonic trends in performance across different electrode configurations. This study highlights the potential of the dimensionality reduction EFSS method as a novel and effective approach for navigation in minimally invasive intracranial surgeries.

## 1. Introduction

Puncture-based brain surgeries are among the most commonly performed procedures for treating various neurological disorders, including brain tumors, hydrocephalus, movement disorders, and epilepsy. These surgeries generally involve inserting specialized surgical instruments such as needles, stimulating electrodes, or ablative electrodes into specific intracranial regions to achieve therapeutic or diagnostic objectives. The accuracy of the instrument placement is critical for the success of these procedures, as improper targeting can lead to complications or reduced therapeutic efficacy. For instance, in brain tumor biopsies, precise targeting of the lesion while avoiding healthy tissue is essential to minimize complications. Similarly, in deep brain stimulation (DBS) surgery, the precise placement of the electrodes in specific brain regions significantly affects the treatment outcomes. Studies have shown that a positional deviation of just 1 mm in DBS electrode placement can reduce the therapeutic effectiveness by over 30% and may result in severe side effects [1].

Puncture-based surgeries have become widely adopted for several conditions due to their minimally invasive nature and proven effectiveness [2,3]. Globally, approximately 100,000 brain tumor biopsies are performed annually, with more than 75% relying on puncture techniques for accurate tissue sampling [4]. DBS procedures, considered the gold standard for treating movement disorders such as Parkinson’s disease, are performed on over 150,000 patients each year [5]. Similarly, hydrocephalus management often involves shunt placement, where precise puncture techniques are critical for catheter insertion and cerebrospinal fluid drainage [6,7].

Recent advancements in medical imaging have also demonstrated the potential of reducing three-dimensional (3D) imaging data to two-dimensional (2D) representations for improving accuracy and computational efficiency. Dimensionality reduction techniques, such as 3D convolutional autoencoders, nonlinear mappings, and distance-preserving algorithms, allow for the projection of high-dimensional medical image data onto 2D spaces while preserving essential features [8,9]. For example, dimensionality reduction has been used to enhance the visualization and registration of complex 3D brain MRI datasets, facilitating a more accurate localization of the surgical targets [10,11]. Additionally, these techniques improve image processing efficiency, enabling faster and more reliable intraoperative navigation [12]. By simplifying complex 3D data into actionable 2D representations, these approaches have the potential to revolutionize imaging-guided surgeries, thereby reducing errors and optimizing procedural outcomes.

Despite their prevalence, these procedures face significant challenges, particularly in real-time localization and accurate navigation. Currently, preoperative imaging modalities such as magnetic resonance imaging (MRI) and computed tomography (CT) are widely used to determine the optimal entry points and trajectories for surgical instruments [13,14]. However, these methods provide only static, preoperative information and cannot account for potential intraoperative changes. To address these limitations, various intraoperative localization techniques have been developed. X-ray imaging is a commonly employed modality, providing intermittent two-dimensional guidance during surgery. However, this method requires acquiring planar images from multiple directions and at different time points, thereby exposing patients and surgeons to high radiation doses, and lacks the ability to provide continuous positional feedback [15,16,17,18]. Stereotactic techniques, both frame-based and frameless, offer high precision for preoperative planning and intraoperative angular adjustments but are inherently limited to preoperative imaging data and often require X-ray assistance for intraoperative localization. Due to the lack of a real-time positioning system, repeated positional adjustments are necessary, which significantly prolongs the duration of the surgeries [19,20,21,22,23]. Ultrasound imaging can provide real-time feedback during surgery, but its application in brain surgeries is restricted due to the attenuation of the ultrasound waves by the skull [24,25,26]. Intraoperative CT and MRI can deliver high-resolution three-dimensional imaging but are often infeasible due to their bulky equipment, operational complexity, and high cost, making them rare in routine surgeries [8]. Magnetic field navigation systems offer real-time three-dimensional tracking but face barriers such as high equipment costs, sensitivity to environmental interference, and limited accessibility in many surgical settings [27,28,29].

To overcome these challenges, this study proposes a novel localization method based on dimensionality reduction electric field source seeking (EFSS). The schematic diagram is shown in Figure 1. The EFSS method enables the real-time localization of surgical instruments, such as electrodes used in puncture-based surgeries, by repurposing the surgical electrodes as localization electrodes. By injecting currents through these electrodes and measuring the surface potentials with electrodes placed on the skull, the system calculates the three-dimensional position of the surgical tool. Compared to conventional X-ray-based positioning methods—which expose both patients and surgeons to significant radiation—our approach relies solely on a simple electric field to position internal surgical instruments. This technique not only effectively reduces radiation exposure but also eliminates the need for complex or bulky equipment and provides a cost-effective and efficient alternative for intraoperative navigation. This study explores electrode configuration optimization and analyzes the factors influencing localization accuracy, ultimately improving the performance of EFSS. Furthermore, the dimensionality reduction method employed in our study maps three-dimensional spatial coordinates onto a two-dimensional plane of interest. This transformation enhances localization accuracy within the target region during puncture procedures, simplifies the model, and provides a more intuitive real-time visualization of relative positions. The proposed method is validated through simulations and experiments on a head phantom, demonstrating promising results. This novel approach provides new insights into real-time navigation for puncture-based brain surgeries, offering a potential solution to longstanding challenges in surgical localization.

## 2. Materials and Methods

### 2.1. Electric Field Source Seeking Method

EFSS is a technique for localizing surgical instruments within the body using an electric field. The method involves injecting a safe current into the body through a localization electrode at the tip of the surgical instrument. This injection generates an electric field within the domain, resulting in a non-uniform potential distribution influenced by the current distribution and the tissue’s conductivity. In this study, the conductivity distribution was assumed to remain constant during surgery, allowing the analysis to focus solely on the impact of the current distribution.

The newly proposed EFSS method shares similarities with Electrical Impedance Tomography (EIT). While in EIT images the impedance distribution of an entire region inherently suffers from low imaging resolution EFSS focuses exclusively on localizing internal current sources. This targeted constraint narrows the computational domain, reduces the computational load, and enhances localization accuracy.

When the position of the localization electrode changes, the resulting alteration in the electric field distribution leads to changes in the potential distribution. Due to the uniqueness theorem of the electric field, there exists a one-to-one correspondence between the current distribution and the potential distribution. Consequently, the current distribution can be deduced from the measured potential distribution, enabling the precise localization of the current source.

To achieve this, a 1 kHz low-frequency sinusoidal current with constant amplitude was employed. At such low frequencies, the electric field can be treated as quasi-static, and dynamic effects can be neglected. Derived from Maxwell’s equations [30], the governing equation for the human body’s computational domain can be expressed as:(1)$$\nabla \sigma \cdot \nabla V+\sigma \nabla^{2}V=0$$
where σ is the conductivity distribution, and V is the electric field distribution in the domain. However, directly obtaining the current distribution from the potential distribution is challenging due to the absence of an explicit relationship. To address this, an iterative method combining forward and inverse problem-solving is typically adopted to approximate the current distribution.

The forward problem in EFSS entails calculating the potential distribution within the domain based on known current and conductivity distributions. This problem is solved using the finite element method (FEM), which involves meshing the domain and performing numerical computations. The FEM process is well documented and can be implemented following the approach outlined in Reference [31]. In this study, the FEM process is represented as an operator h. The calculation process of FEM can be expressed as(2)φ=h(J,σ)

An initial guess $J_{0}$ for the current distribution is used to compute the corresponding potential at the measurement electrode positions. The difference between the computed potential $\varphi_{0}$ and the actual measured potential $\varphi_{meas}$ is defined as the computational error E.(3)$$E={\left\|\varphi_{meas}-\varphi_{0}\right\|}_{2}$$

By calculating the Jacobian matrix Y of the current vector with respect to the potential, the current vector is iteratively updated(4)$$Y={\left.\frac{\partial \varphi (J)}{\partial J}\right|}_{J=J_{0}}$$

The iterative process can be expressed as(5)$$J^{k+1}=J^{k}-{\left(Y_{k}^{T}Y_{k}\right)}^{-1}Y_{k}^{T}\left[h(J^{k},\sigma )-\varphi_{meas}\right]$$

This iterative forward-inverse problem-solving process continues until the computational error falls below a predefined threshold. The final estimated current vector represents the least-squares solution for the current distribution.

### 2.2. Dimensionality Reduction

While a three-dimensional (3D) electrode configuration allows for the localization of an object in 3D space, achieving the high precision required for brain surgical navigation remains challenging. This limitation is particularly significant given the stringent accuracy demands of brain surgeries. In the case of puncture-based procedures, preoperative imaging techniques such as CT or MRI are used to design a surgical path that avoids critical structures and blood vessels. During the procedure, the surgical instrument is navigated and positioned along this predefined path. Consequently, reducing instrument localization to two dimensions (2D) can effectively improve precision.

To enhance localization accuracy for objects constrained to move within a specific plane, a dimension reduction technique is employed. The original 3D coordinates are projected onto a plane that intersects the predefined path for more accurate localization. However, since infinitely many planes can pass through the predefined path, it is crucial to determine the specific plane equation and map the 3D coordinates onto the selected 2D plane. This transformation is performed by rotating the coordinate system around a predefined axis and projecting the target point onto the resulting 2D plane.

The dimensionality reduction method narrows the overall three-dimensional localization to a two-dimensional region of interest, thereby reducing the error associated with an extra dimension and enhancing localization precision. Additionally, this approach simplifies the model and reduces the computational load, which allows for finer discretization of the computational grid and further improves the localization accuracy.

In the computational domain Ω, the original 3D coordinate system {x,y,z} is first established. Let the predefined surgical path be represented by the line segment *PQ*, where *P* denotes the entry point, and *Q* denotes the target point, with $P={\left(x_{P},y_{P},z_{P}\right)}^{T}$, and $Q={\left(x_{Q},y_{Q},z_{Q}\right)}^{T}$. The direction vector d of the path *PQ* is calculated as follows:(6)d=Q−P

The normalized direction vector is(7)$$d_{unit}=\frac{d}{\left\|d\right\|}$$

Assuming that the positioning coordinates in three-dimensional space are $X={\left(x,y,z\right)}^{T}$, to rotate a point X by the angle θ the axis defined by points *P* and *Q*, the coordinate system is rotated around the *PQ* axis to obtain the rotated coordinate system {u,v,w}.

At first, shift the origin to P(8)$$X_{shifted}=X-P$$
and then rotate the global coordinate system such that the $z^{\prime }$-axis aligns with *PQ*. define the rotation matrix T for this transformation:(9)$$T=\left[\begin{array}{c} e_{x^{\prime }}^{T}\\ e_{y^{\prime }}^{T}\\ e_{z^{\prime }}^{T} \end{array}\right]$$
where $e_{z^{\prime }}=d_{unit}$ is orthogonal to $e_{x^{\prime }}$, and $e_{y^{\prime }}$ is orthogonal to both $e_{x^{\prime }}$ and $e_{z^{\prime }}$(10)$$e_{x^{\prime }}=\frac{e_{x}-(e_{x}\cdot e_{z^{\prime }})e_{z^{\prime }}}{\left\|e_{x}-(e_{x}\cdot e_{z^{\prime }})e_{z^{\prime }}\right\|}$$(11)$$e_{y^{\prime }}=e_{z^{\prime }}\times e_{x^{\prime }}$$

Apply a rotation of θ about the $z^{\prime }$-axis using:(12)$$R_{\theta }=\left[\begin{array}{ccc} \cos\theta & -\sin\theta & 0\\ \sin\theta & \cos\theta & 0\\ 0 & 0 & 1 \end{array}\right]$$

Finally, return to the original coordinate system:(13)$$X_{\mathrm{rotated}}=T^{T}\cdot (R_{\theta }\cdot (T\cdot X_{\mathrm{shifted}}))+P$$

The resulting coordinates of the rotated point are $X_{rotated}={\left(u,v,w\right)}^{T}$.

### 2.3. Electrode Arrangement Optimization

In the process of two-dimensional (2D) localization of surgical instruments during operations, determining the reference plane for the surgical path is a critical step, as there are infinitely many possible planes that can be chosen. The arrangement of the electrodes significantly influences the localization accuracy of the EFSS method. Therefore, preoperative evaluation metrics are necessary to assess electrode arrangements and guide the selection of an optimal configuration to improve localization precision.

Due to the inherent similarity between EFSS and EIT, their primary distinction lies in their objectives. In EIT, many studies adopt the sensitivity of externally measured potential variations—caused by conductivity changes—as an optimization index for the electrodes [32]. In contrast, in the EFSS method, when the position of the localization electrode changes, the sensitivity of the measured potential variation reflects its ability to detect these positional changes. Drawing on the concept of sensitivity, we propose an electrode optimization metric tailored for EFSS to quantitatively assess the electrode’s responsiveness to positional variations.

Therefore, two performance metrics, Max Relative Change (MRC) and Mean Relative Change (MRC-mean), are introduced to optimize the electrode arrangement for 2D EFSS localization.

The MRC quantifies the relative magnitude of the maximum potential variation. A higher MRC value indicates greater sensitivity of the electrodes to positional changes, making it a reliable metric for optimization. Its formulation is expressed as(14)$$\mathrm{MRC}=\frac{\max(|\varphi_{i}-\varphi_{j}|)}{\max(\varphi_{i},\varphi_{j})}$$
where $\varphi_{i}$ is the measured potential value of the corresponding electrode before localization electrode movement, and $\varphi_{j}$ is the measured potential value of the corresponding electrode after localization electrode movement.

On the other hand, the MRC-mean measures the average relative intensity of potential variations across the electrode array. This metric addresses the potential numerical instability of the MRC in specific scenarios. A higher MRC-mean value reflects more significant overall potential changes, indicating a higher average sensitivity to electrode position variations. It is expressed as:(15)$$\mathrm{MRC}\text{-}\mathrm{Mean}=\frac{1}{N}\overset{N}{\underset{i=1}{\sum }}\frac{|\varphi_{i}-\varphi_{j}|}{\max(\varphi_{i},\varphi_{j})}$$
where *N* is the total number of steps required for the entire electrode movement process.

By integrating these two performance metrics, the optimization and selection of the electrode array arrangement were achieved, ensuring improved localization accuracy and robustness in 2D EFSS applications.

### 2.4. Localization System

The surgical localization system (Figure 2) utilized in the EFSS method comprises four main components: a constant-current source module, a cyclic potential acquisition module, an EFSS localization host computer, and a mechanical motion module. Its specific technical specifications are shown in Table 1.

The constant-current source module integrates a localization electrode located at the front end of the simulated surgical instrument and a constant-amplitude current source. The localization electrode used in our study had a diameter of 0.7 mm. The electrode needle was fabricated from SUS304 stainless steel, with its rear end connected to the constant-current source via a fine guide wire. Given its small size, it was typically treated as a point electrode during localization. Its primary function was to inject a 1 mA 1 kHz constant-amplitude sinusoidal current into human tissue via the localization electrode, ensuring stable current delivery during the procedure.

The outer shell was fabricated using resin via 3D printing, with its shape constructed based on cranial dimensions. The inner dimensions measured 220 mm × 200 mm × 100 mm, and a wall thickness of 8 mm was employed to secure the measurement electrodes. Fifteen measurement electrodes were fixed on the outer shell, and their positions could be adjusted according to the localization requirements. The electrode tips were made of copper, with their rear ends connected to the cyclic potential acquisition module via fine guide wires. This module includes potential measurement electrodes arranged on the surface of the human body and a multi-channel selection switch. This module collects the potential distribution generated on the body surface as a result of the electric field formed within the tissue. The acquired potential data are then transmitted to the EFSS localization host computer, which employs the EFSS method to precisely localize the localization electrode and compute its coordinates.

The mechanical motion module comprises a robotic arm and a stepper motor. The robotic arm facilitates angle adjustment and guides the initial entry point, while the stepper motor ensures the stable and high-precision linear insertion of the surgical instrument. The robotic arm is capable of rotating freely within a 180° range above the cranial shell. Additionally, the stepper motor provides an effective travel of 200 mm, which is sufficient to meet the requirements for electrode implantation in brain surgery. This module enables the surgical instrument to be deployed into the surgical region based on predefined initial coordinates and orientation. The system achieves exceptional precision, with angular errors controlled within 0.1°, and depth errors within 0.07 mm. These stringent tolerances ensure a reliable angle and depth accuracy during motion. Consequently, in the subsequent simulated surgical procedures, this module primarily served as a standard reference coordinate system for validating the calculated localization coordinates.

## 3. Results

### 3.1. Simulation

To validate the effectiveness of the proposed method, simulations were designed. The model dimensions were based on standard human head measurements and subsequently refined using smoothing techniques, with dimensions of 220 mm × 200 mm × 100 mm, as shown in Figure 3a. In the simulation, the COMSOL 6.1 Electric Currents module was employed; a constant-current source was implemented via a pair of point current sources with opposite polarities, and electrode 16 was designated as the ground. A refined mesh was applied within 1 mm of the localization plane, with a maximum element size of 0.0177 m and a minimum element size of 0.001 m, while a coarser mesh was used for the remaining regions, featuring a maximum element size of 0.0222 m and a minimum element size of 0.00399 m. Owing to the simple electric field involved in the EFSS method, the entire computation could be completed within 10 s on a standard PC. To replicate the linear motion of a surgical instrument along a predefined path during puncture procedures, the line segment PQ was designated as the planned surgical trajectory, with point P marking the surgical entry point, and point Q representing the surgical target, as illustrated in Figure 3b. The current injection point was positioned along the PQ line segment, the current exit point was assigned to one of the surface electrodes, and the grounding point was set at the reference electrode (R point). During the simulation and the experiment, electrode 16 was always set as the ground electrode (shown in Figure 4).

The effect of the skull thickness on the electric field was not considered in this simulation model. This is because, in actual surgical procedures, CT or MRI imaging of the patient’s head can be performed preoperatively to create a refined three-dimensional model. This can explain the potential distribution after considering the influence of the human skull on the electric field in actual situations.

Furthermore, four 2D localization planes intersecting the PQ line segment at 30° intervals were defined and named Plane 00, Plane 30, Plane 60, and Plane 90. A top view of the electrode arrangement with respect to the four planes is shown in Figure 4. The planes from 120° to 180° were not considered because these planes were completely symmetrical to the previous planes, and the calculation results were the same. The measurement electrodes were placed as close as possible to these planes, forming the simulation model as shown in Figure 3d. Due to the model’s bilateral symmetry, electrode arrangements within the range of 0–180° are representative, and a 30° interval yielded stable optimization indices with minimal sensitivity to measurement errors. Moreover, smaller intervals resulted in practical issues, such as overlapping fixed holes on the outer shell, making the interval of 30° an optimal balance for simulation validation. The black dots in the illustration indicate the positions of the localization electrodes. This model was primarily developed to provide a computational framework for solving the forward problem.

During meshing, a refinement strategy was adopted to enhance the localization resolution while minimizing the total number of meshes. Since the localization region of interest was near the localization plane, we refined the mesh within a 1 mm distance from the plane, using a minimum mesh size of 1 mm; meanwhile, other regions were meshed more coarsely, with a minimum size of 4 mm, yielding the meshing result shown in Figure 3c.

To simulate the movement of the localization electrodes during puncture procedures, the minimum step size for electrode displacement was set to 1 mm. A total of 92 localization electrode points were defined along the PQ line segment from point P to point Q. To optimize the electrode arrangement, the measured potentials of these 92 points were calculated, and then the evaluation metrics based on the measured potentials were computed.

In the simulation, the conductivity of the brain material was set to 0.225 S/m, and the current source supplied a 1 kHz, 1 mA constant current, which is safe for brain surgery [33].

#### Simulation Result

Two performance metrics were calculated for the four planes, with the results presented in Table 2. Both the MRC and the MRC-mean metrics exhibited a monotonically increasing trend. Higher values of these indicators indicate a greater sensitivity to changes, meaning they respond more significantly to variations. Thus, based on the simulation results, the localization performance followed the order: Plane 90 > Plane 60 > Plane 30 > Plane 00. Consequently, Plane 90 was identified as the optimal electrode arrangement.

### 3.2. Experiment

The simulation-derived ranking of the four planes, based on the evaluation metrics, confirmed Plane 90 as the optimal configuration for electrode placement. However, further experimental validation was required to corroborate this finding. The experimental flowchart is shown in Figure 5. The overall system workflow begins with the potential acquisition of external voltage data while the surgical instrument is at its initial position. The acquired signal is then filtered, amplified, and processed using fast Fourier transform sampling in the pre-processing stage to yield a 1 kHz signal. Next, the EFSS method is applied for localization calculation, and the resulting localization is visualized to assess whether the predetermined final target point has been reached. If any angular or depth deviations are detected, the Mechanical Motion Module is employed to make the necessary adjustments, and the localization process is repeated until the final target is achieved.

#### 3.2.1. Simulated Surgery Setup

The interior of the brain model was filled with a NaCl solution adjusted to emulate the electrical conductivity of real human brain tissue. The brain tissue is mainly composed of white matter and gray matter, whose conductivities are sufficiently similar to be approximated as identical. The brain tissue was simulated using an NaCl solution—a widely accepted method—which allowed for the precise adjustment of conductivity through a controlled NaCl concentration. A conductivity meter was employed to verify that the solution maintained a conductivity of 0.225 S/m, consistent with the simulation parameters. During the experiment, a 1 kHz, 1 mA sinusoidal current was supplied by the constant-current source.

In the EFSS framework, potential acquisition was implemented by maintaining a constant localization electrode for current injection while designating electrode 16 as the fixed reference. The current return electrode was systematically varied from electrode 2 to 15, and for each injection–return pairing, potential measurements were acquired from the remaining measurement electrodes relative to the reference. All obtained potential data were then recorded, resulting in a dataset comprising 182 distinct data points.

To assess the impact of the four different electrode arrangements on localization performance and validate their alignment with the proposed evaluation metrics, a series of experiments were conducted. The actual picture of the electrode arrangements with respect to the four planes is shown in Figure 6. To simulate a realistic puncture procedure, a hole was created on the fixed electrode shell to serve as the electrode entry point (designated as point P(0, 79.90, 131.60) (mm)). An internal point, Q(0, 0, 85.50) (mm), was selected as the target point for the experiment. During the procedure, the mechanical motion module was configured with the angle and entry point coordinates corresponding to the PQ trajectory. The surgical instrument, driven by a stepper motor, moved incrementally along the predefined path PQ. Throughout the motion, a constant alternating current was continuously injected via the localization electrode positioned at the instrument’s tip, and the EFSS method was employed to perform real-time localization. The localization results were subsequently compared with the actual PQ trajectory to evaluate the procedure accuracy.

For consistency, the EFSS localization was conducted considering the same PQ path using the four distinct electrode arrangements. Each electrode arrangement was tested five times to account for random variations and minimize the impact of experimental errors.

#### 3.2.2. Feasibility Analysis Results

To verify the EFSS method’s capability to achieve accurate and unique solutions for 2D localization, the localization electrode was placed at the target point Q. In the 2D plane, all points were hypothetically treated as current injection points. The potential differences between simulated and measured electrode potentials were computed, generating error distribution maps. This process was repeated for all four planes, and the results are illustrated in Figure 7. The data were visualized in both the original and the adjusted coordinate systems to improve clarity. In the unadjusted side view, the results showed distortion, which hindered the precise observation of relative positions. After coordinate adjustment, the relative changes in the localization point within the plane became more apparent.

The red ‘x’ in Figure 7 denotes the position with the lowest error. Across all cases, the error decreased monotonically from high-error regions at the periphery to the lowest-error point, forming a convex function with a single global minimum. This characteristic confirmed the method’s effectiveness and uniqueness, making it well suited for providing the optimal solution via the gradient descent method described in Section 2.1.

The distance errors between the coordinates of the minimum-error point and those of the target point Q are summarized in Table 3. For each plane, five repeated experiments were conducted to minimize the impact of random errors. The results demonstrated that the errors across all planes remained within 3 mm, proving the effectiveness of single-point localization.

#### 3.2.3. Localization Result

A simulated experiment was conducted to evaluate electrode localization along the P-to-Q path across the four planes, with one setup shown in Figure 7. And the real position of the localization electrode during the experiment is shown in Figure 8. The yellow arrows indicate the actual position of the localization electrode. Each plane was tested five times to ensure repeatability, and the distance errors from the localization experiments are displayed in Figure 9. Most errors remained within 2.5 mm, with only a few exceptions exceeding this value. The trends across repeated experiments were consistent, highlighting the method’s robustness against system and measurement errors.

The average distance error and standard deviation for each plane were also computed, as shown in Table 4. The average distance error for all planes was below 2 mm. The ranking of the average errors was consistent with the recommended electrode arrangement order based on the evaluation metrics, as follows: Plane 90 < Plane 60 < Plane 30 < Plane 00. These findings validated that the proposed evaluation metrics can effectively guide electrode arrangement optimization for EFSS-based puncture surgeries. After electrode optimization, the average localization error was reduced by 56%.

### 3.3. Discussion

The simulation and experimental results demonstrated that the EFSS method is effective for single-point localization and straight-path localization in puncture surgeries. The dimensionality reduction approach, which reduces 3D localization to 2D, consistently achieved an average localization error within 2 mm, confirming the method’s reliability. This level of accuracy is only slightly lower than that of the most widely used frame-based positioning methods and essentially meets the precision requirements for brain surgery [27,29].

Furthermore, the evaluation metrics (MRC and MRC-mean) and localization errors exhibited monotonic trends as the planes transitioned from vertical to flatter orientations. For puncture surgeries, the proposed metrics can effectively guide electrode arrangement optimization. The results indicate that arranging electrodes on both sides of the puncture path to better enclose it yields superior localization accuracy.

However, this study only evaluated electrode arrangements at specific angles in 2D planes due to experimental constraints. Since it was not feasible to place electrodes at all possible positions, validation was limited to discrete angles. Future work will explore more general scenarios using alternative methods.

## 4. Conclusions

This study proposes a dimensionality reduction electric field source seeking (EFSS) surgical navigation method for real-time, high-precision instrument localization in intracranial puncture surgeries. The localization system integrates internal localization electrodes and external potential measurement electrodes, minimizing surgical trauma. Two evaluation metrics, MRC and MRC-mean, were introduced to optimize the electrode arrangement preoperatively through simulations, enabling improved localization performance. Both simulations and experiments demonstrated that the dimensionality reduction EFSS method consistently maintained an average localization error within 2 mm. The evaluation metrics effectively guided electrode arrangement optimization, reducing the localization error by 56% post-optimization. This method presents a novel pathway for advancing intracranial surgical navigation techniques.

## Figures and Tables

**Figure 1 sensors-25-01378-f001:**
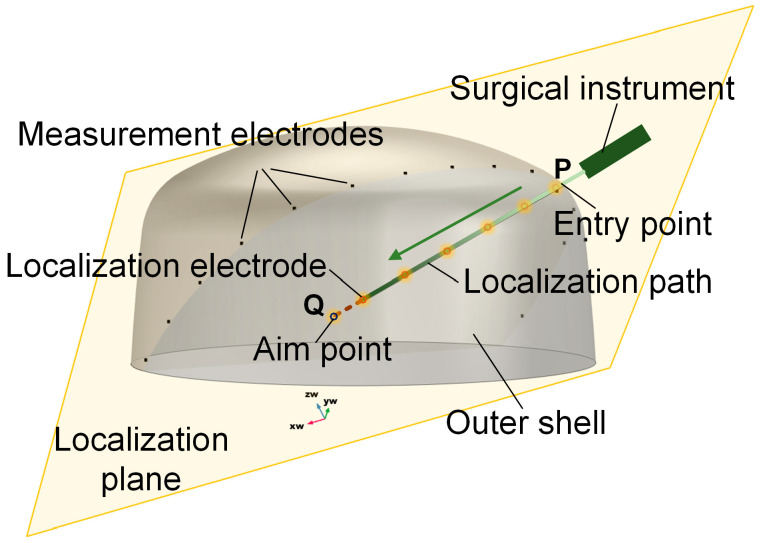
Schematic diagram of EFSS. The yellow plane is the localization plane.

**Figure 2 sensors-25-01378-f002:**
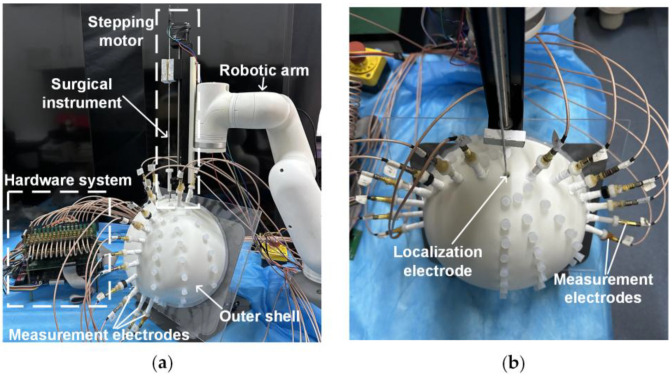
Localization system. (**a**) Side view. (**b**) Top view.

**Figure 3 sensors-25-01378-f003:**
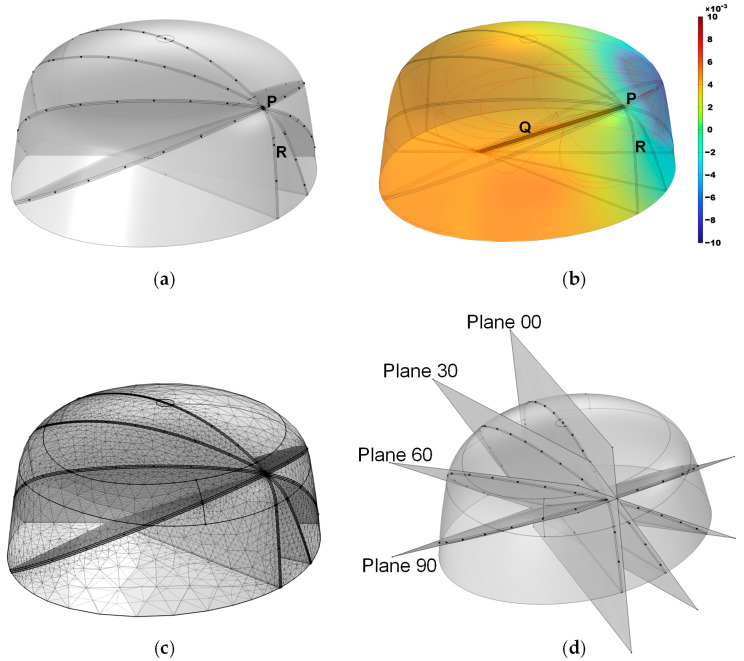
Simulation model. (**a**) Overall structure, where the black dots indicate potential positions for the external electrodes. (**b**) Simulated potential distribution, with the color bar representing potential magnitude. (**c**) Mesh discretization. (**d**) Positions of the four planes.

**Figure 4 sensors-25-01378-f004:**
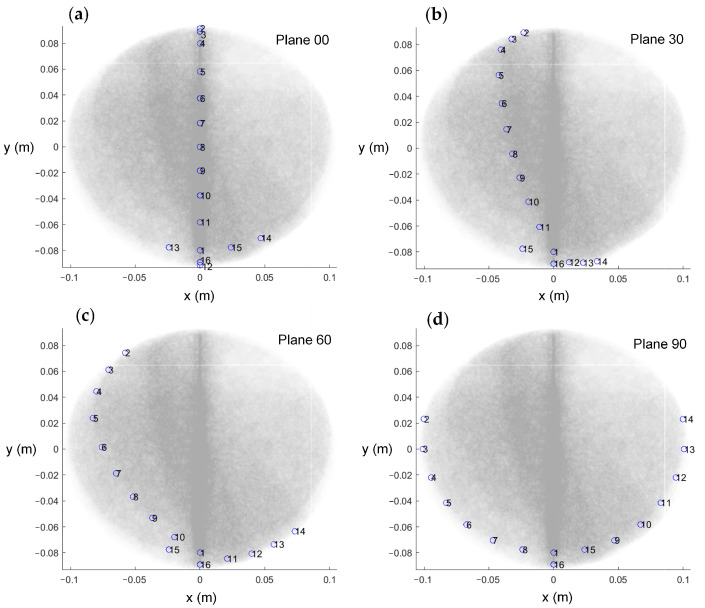
The top view of the electrode arrangement with respect to the four planes. (**a**) Plane 00. (**b**) Plane 30. (**c**) Plane 60. (**d**) Plane 90.

**Figure 5 sensors-25-01378-f005:**
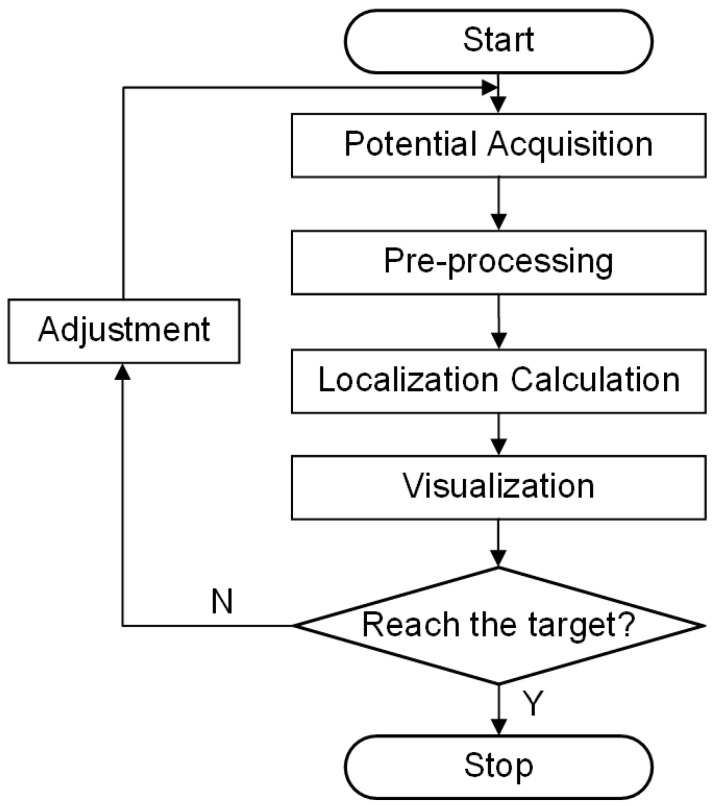
Experimental flowchart.

**Figure 6 sensors-25-01378-f006:**
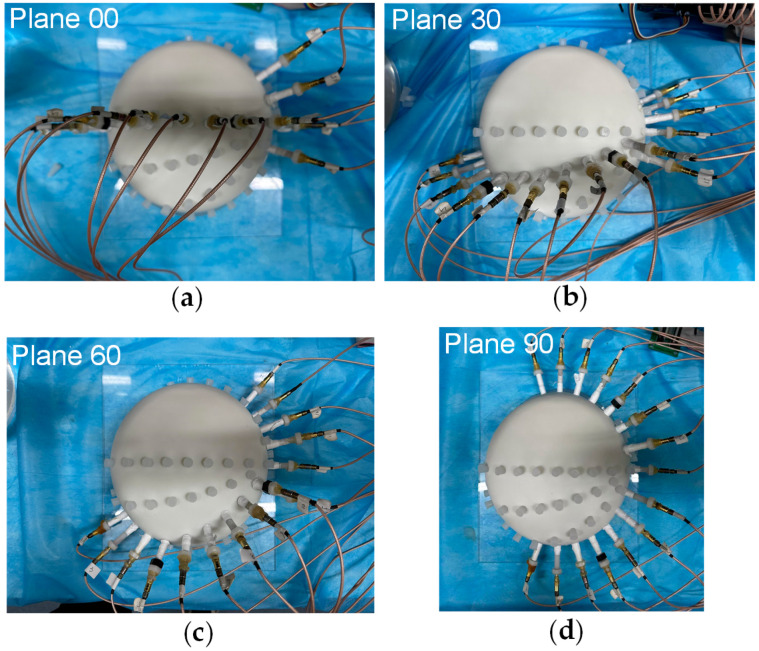
Actual picture of electrodes arrangement with respect to the 4 planes. (**a**) Plane 00. (**b**) Plane 30. (**c**) Plane 60. (**d**) Plane 90.

**Figure 7 sensors-25-01378-f007:**
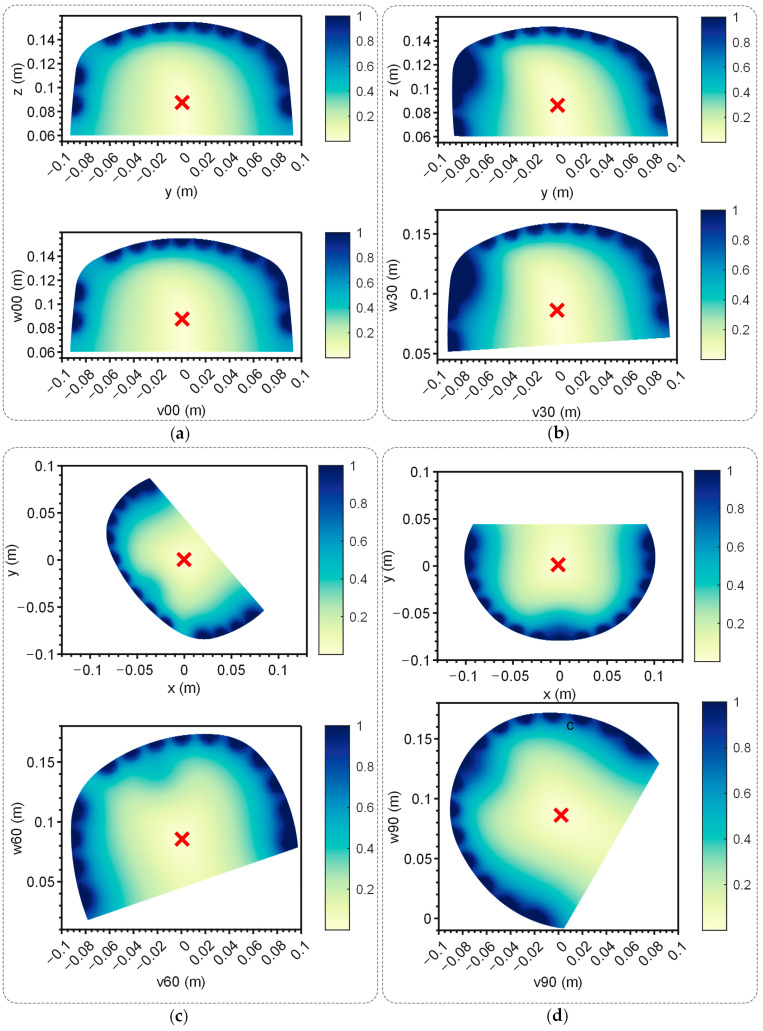
Error map of the 4 planes. Up: Error map of the yoz plane when the electrodes were distributed on the corresponding plane. Down: Error map of the yoz plane when the electrodes were distributed on corresponding plane after coordinate transformation. The red ‘x’ indicates the lowest point. (**a**) Plane 00. (**b**) Plane 30. (**c**) Plane 60. (**d**) Plane 90.

**Figure 8 sensors-25-01378-f008:**
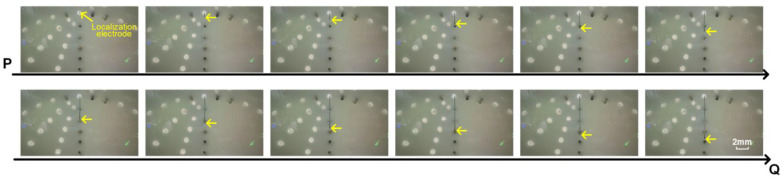
The real position of the localization electrode during the experiment. The white circles indicate the locations where the measurement electrode holes were not arranged. The black circles indicate the measurement electrodes. The yellow arrows indicate the actual position of the localization electrode.

**Figure 9 sensors-25-01378-f009:**
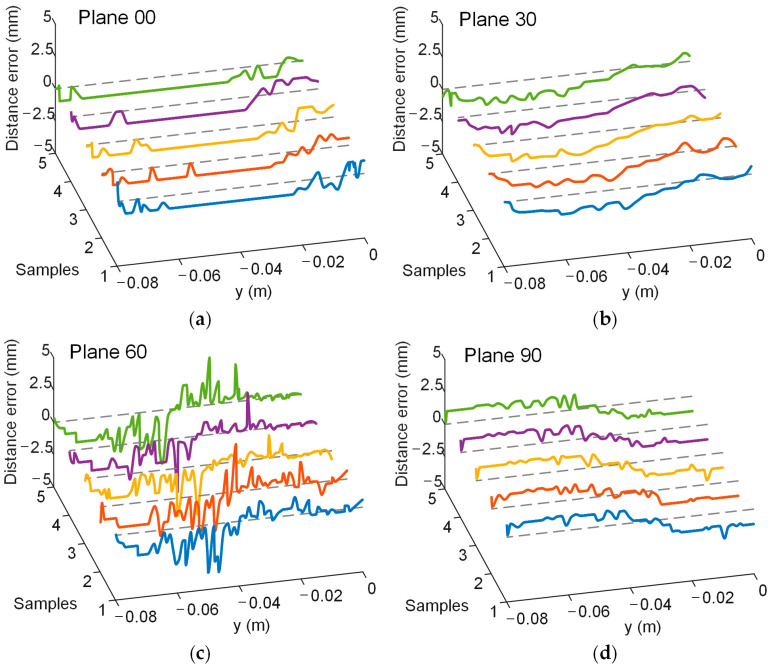
Distance error in the experiment. Different colored curves represent the results of multiple repeated experiments. The gray dashed line represents the position where the distance error is 0. Lines in five different colors indicate the results of five repeated experiments. (**a**) Plane 00. (**b**) Plane 30. (**c**) Plane 60. (**d**) Plane 90.

**Table 1 sensors-25-01378-t001:** Technical specifications of the localization system components.

Component	Parameter	Specification/Value
Constant-Current Source Module	Amplitude	1 mA
	Frequency	1 kHz
	Waveform	Constant-amplitude sinusoidal
Localization Electrode	Material	SUS304 stainless steel
	Diameter	0.7 mm
	Connection	Fine guide wire
Cyclic Potential Acquisition Module	Outer shell material	Resin
	Number of measurement electrodes	15
	Electrode tip material	Copper
	Channel count	16
	Acquisition speed	100 kHz
	ADC bit resolution	12
	Acquisition accuracy	1 mV
Mechanical Motion Module	Rotational range	180° above cranial shell
	Angular accuracy	0.1°
	Effective travel	200 mm
	Depth accuracy	0.07 mm

**Table 2 sensors-25-01378-t002:** Evaluation metrics results of the 4 electrode arrangement planes.

Metrics	Plane 00	Plane 30	Plane 60	Plane 90
MRC	0.555	0.582	0.684	0.697
MRC-mean	0.077	0.088	0.091	0.093

**Table 3 sensors-25-01378-t003:** The error between the lowest point of the error distribution and the target point.

	Plane 00	Plane 30	Plane 60	Plane 90
Average distance error (mm)	2.12	0.74	0.56	2.52
Standard deviation (mm)	1.25	0.58	0.44	0.95

**Table 4 sensors-25-01378-t004:** Localization error for the 4 planes.

	Plane 00	Plane 30	Plane 60	Plane 90
Average distance error (mm)	1.84	1.59	1.00	0.82
Standard deviation (mm)	1.05	1.30	0.98	0.37

## Data Availability

The data presented in this study are available on request from the corresponding author.

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
