# Peer review of "Dimensionality Reduction and Electrode Arrangement Optimization for an Electric Field Source Seeking Surgical Navigation Method"

_sensors, 2025, doi:10.3390/s25051378_

Round 1
Reviewer 1 Report
Comments and Suggestions for Authors
In this paper, Fang et al., reported a feasible method to position the implantable needle-type electrode during the surgery. Authors provided clear background, methods, and results. Here are comments the authors would consider.
1. Is there any other electric field-based electrode positioning methods in the literature? If so, what is the novelty of this manuscript compare to them?
2. What is the minimum size (diameter) of the needle-type electrode that this methods can detect? what is the resolution of this method at the tip of the needle electrode?
3. Is there any effect of electrical current flows through the electrode? What is the minimum amount of current that the system can detect?
4. When applying the current to the electrode, where is the ground? If the current flows through brain to the sensing electrode arrays, did author also consider the safety issue?
5. Authors reported the error of this system is within 2 mm. Is it accurate? The authors should provide the accuracy of other imaging-based methods. Furthermore, is this accuracy enough for positioning of electrode during the surgery? Please clarify this.
6. In Figure 7, what are white and black circles in figures? Also please add the scale bars.
Reviewer 2 Report
Comments and Suggestions for Authors
The paper proposes a localization method based on reduced-dimensional electric field source seeking. The proposed study seems interesting; however, several issues suggest that the paper should be rewritten and completed, filling methodological gaps. The main concerns are outlined below.
INTRODUCTION SECTION
- The whole section requires additional supporting references.
- The novelty of the proposed method is unclear. Lines 81 to 93 refer to the EFSS method, but it is not clear in this paragraph what the novelty contribution is compared to the current literature. Has the EFSS method already been proposed in the literature? In which applications? What about the dimensionality reduction applied to it? Limitations and disadvantages of existing solutions should be better outlined to highlight the aim and novelty of this work.
MATERIALS AND METHODS SECTION
- Subsections 2.1 and 2.2 lack supporting references. Again, the novelty contribution is unclear in the description of both the EFSS method and dimensionality reduction. In addition, a large part of these subsections refers to well-known relationships and details (e.g., Ohm’s law) that could be removed by citing them from the literature.
- Figure 1 is not cited and described in the text.
- Subsection 2.3 also lacks supporting references. In what applications have the max relative change and the mean relative change been used? Why were these metrics chosen?
- The surgical system in subsection 2.4 is described poorly and qualitatively. Details and technical specifications of the system components are lacking.
- What about the calibration procedure?
RESULTS SECTION
- Again, the simulation in subsection 3.1 is described poorly and qualitatively. What software was used to run this simulation? With what settings and at what computational burden? The dimensions of the implemented model should be made explicit.
- Figure 3a is not cited and described in the text.
- It is unclear why a 30° interval was chosen in the selection and testing of localization planes. Was this interval deduced from the scientific literature?
- The refinement strategy adopted to enhance localization resolution deserves to be described in detail.
- How was the monotonically increasing trend in the results in Table 1 determined?
- The simulated surgical setup in subsection 3.2.1 is described poorly and qualitatively. Details and technical specifications are lacking. How was the brain model made? With what material and what dimensions? How was it verified that the conductivity was “precisely set to 0.225 S/m”?
- The acquisition protocol is poorly How many tests and acquisitions have been carried out? In which conditions? A proper description of the acquisition protocol is missing.
- Data processing is not clearly outlined. I also suggest adding a flowchart or a block diagram showing the main processing steps.
- The distance errors in Table 2 cannot be compared without measurement uncertainty.
- The number of digits used to express the results in Tables 2 and 3 is wrong.
Round 2
Reviewer 1 Report
Comments and Suggestions for Authors
The authors addressed the reviewer's comments accordingly. I think this manuscript is ready to be published.
Author Response
Thank you very much for your positive feedback and thoughtful review. We truly appreciate your guidance throughout the revision process and are delighted that our responses have met your expectations.
Reviewer 2 Report
Comments and Suggestions for Authors
I appreciate the efforts to improve the manuscript. Although the authors have fixed most of the issues raised in the previous revision stage, the following significant issues have yet to be addressed.
- Technical specifications of the surgical localization system components are still lacking. I suggest listing them all in a table with their specifications.
- How was the outer shell of the brain model shown in Figs. 2 and 6 made? With what material and what dimensions?
- Simulation settings and details are not provided. How was the simulation model drawn? Which COMSOL modules are used? What are the mesh settings?
- Details and a proper description of the pre-processing phase shown in Fig. 5 are not provided.
- The number of digits used to express the results in Tables 2 and 3 is still wrong. The significant figures depend on the measurement uncertainty, which should be expressed with one digit unless its value is 1 or 2 (in that case, rounding off to one digit may lead to a significant loss of information). I suggest the following references:
[1] Taylor J.R. "An Introduction to Error Analysis: The Study of Uncertainties in Physical Measurements", 3rd edition, University Science Books, 2022.
[2] GUM: Guide to the Expression of Uncertainty in Measurement (https://www.bipm.org/en/publications/guides/gum.html)
